# *Drosophila* Males Differentially Express Small Proteins Regulating Stem Cell Division Frequency in Response to Mating

**DOI:** 10.3390/jdb13030021

**Published:** 2025-06-23

**Authors:** Manashree S. Malpe, Leon F. McSwain, Heath M. Aston, Karl A. Kudyba, Chun Ng, Megan P. Wright, Cordula Schulz

**Affiliations:** 1Department of Cellular Biology, University of Georgia, Athens, GA 30602, USA; manashree.malpe10@gmail.com (M.S.M.); leon.foy.mcswain@emory.com (L.F.M.); heathaston99@gmail.com (H.M.A.); karlkudyba@gmail.com (K.A.K.); megan.wright1@uga.edu (M.P.W.); 2Avanti Corporation, Alexandria, VA 23315, USA; 3Department of Pediatrics, Emory University, Atlanta, GA 30322, USA; 4Medical College of Georgia, Augusta University, Augusta, GA 30912, USA; 5Oak Ridge Institute for Science and Education, US Department of Energy, Oak Ridge, TN 37830, USA; 6Mass General Brigham Innovation, Summerville, MA 02145, USA

**Keywords:** mating, germline stem cell division frequency, secretion, small proteins, DEGs

## Abstract

The germline stem cells (GSCs) in the male gonad of *Drosophila* can increase their division frequency in response to a demand for more sperm caused by repeated mating. However, the molecules and mechanisms regulating and mediating this response have yet to be fully explored. Here, we present the results of a transcriptome analysis comparing expression from the testis tips from non-mated and mated males. An overlapping set of 18 differentially expressed genes (DEGs) from two independent *wild-type* (*wt*) strains revealed that the majority of the DEGs encode secreted proteins, which suggests roles for them in cell–cell interactions. Consistent with a role for secretion in regulating GSC divisions, knocking down Signal Recognition Particle (SRP) components within the germline cells using RNA Interference (RNAi), prevented the increase in GSC division frequency in response to mating. The major class of DEGs encodes polypeptides below the size of 250 amino acids, also known as small proteins. Upon reducing germline expression of small proteins, males no longer increased GSC division frequency after repeated mating. We hypothesize that mating induces cellular interactions via small proteins to ensure continued GSC divisions for the production of sperm.

## 1. Introduction

The ability to produce gametes is key to an individual’s fitness and to the survival of a species. Studies performed in various model organisms revealed that life-history traits can modify fecundity. For example, when the field cricket, *Teleogryllus oceanicus*, is exposed to acoustic sexual signals during juvenile stages, it develops more reproductive tissue compared to its siblings kept in a silent environment [1]. Frogs and birds show plasticity in their ejaculate depending on the number of other males in the colony [2,3]. Thus, studies have demonstrated that males can adjust their sperm pool to situations of demand.

For most species, sperm production depends on the activity of GSCs. These precursor cells have been intensively studied in many model organisms, including the fruit fly, *Drosophila melanogaster*. In *Drosophila*, the position of the GSCs and their pattern of division are well known. GSCs are located at the tip of the male gonad, where they are attached to post-mitotic somatic hub cells. Next to the GSCs at the hub lie somatic cyst stem cells (CySCs). Two CySCs enclose one GSC in cytoplasmic extensions that extend into the hub cells [4]. Interactions between hub cells, GSCs, and CySCs regulate their association and function [5,6]. For example, signaling via Janus Kinase/Signal Transducer and Activator of Transcription, Bone Morphogenetic Protein, and Hedgehog are required for GSC and CySC fates [7,8,9,10,11,12,13].

When GSCs and CySCs divide, they produce new stem cells that remain attached to the hub and daughter cells destined for differentiation. The daughter cell, destined for differentiation, is displaced away from the hub [14]. This cell, called a gonialblast (GB), becomes enclosed by two cyst cells, thereby forming the cyst, which is the developmental unit within the *Drosophila* gonads [4,15,16]. The enclosed gonialblast amplifies itself via four transit amplifying divisions that produce 16 spermatogonia. Spermatogonia enlarge, reduce their DNA content via two rounds of meiosis, and differentiate into exactly 64 spermatids (Figure 1A) [4,17]. The cyst cells normally do not divide anymore, but grow in size and co-differentiate with the enclosed germline cells [18].

The tight physical association between the germline cells and the cyst cells, as well as continued communication between the two cell types, allow the cells of the cyst to differentiate into later stages [19,20,21]. The cyst cells fail to enclose the germline cells when animals lack signaling via the Epidermal Growth Factor (EGF). The germline cells then proliferate at early stages, forming germline tumors [15]. Notably, loss of EGF signaling also causes the GSCs at the hub to divide at significantly higher frequencies compared to GSCs in control animals. This suggests that germline enclosure is essential for reducing GSC division frequency, possibly by providing a physical barrier for signaling molecules [22].

While the *Drosophila* GSC field has revealed many molecules and mechanisms that guide cell fates and differentiation, much less is known about GSC division frequency. In *Drosophila* males, external signals regulate GSC divisions. For example, GSCs divide less frequently when nutrients are unavailable, and this response is regulated by insulin signaling [23]. Similarly, mating status influences GSC divisions. When *Drosophila* males are repeatedly mated, it causes a significant increase in the division frequency of GSCs, GBs, and spermatogonia [24,25]. The increase in GSC division frequency depends on the activity of seven non-redundant G-protein coupled receptors [24]. However, how these GPCRs are activated and how they regulate GSC divisions remains elusive.

We used a transcriptome analysis to further explore the molecules involved in the increase in GSC division frequency in response to mating and to potentially fill some gaps in our understanding. Our analysis revealed 18 DEGs in response to mating. Among the DEGs with the most significant change in response to mating, all but one encode potentially secreted proteins, and most of these are small proteins. Small proteins are polypeptides below the size of 250 amino acids. They have a variety of functions in plant and animal species, including signal transduction, transcription, and RNA metabolism [26]. Here, we show that impairing secretion or small protein expression blocks the increase in GSC division frequency in response to mating.

## 2. Methods

### 2.1. Fly Husbandry

Flies were raised on a cornmeal/agar diet in temperature-, light-, and humidity-controlled incubators. Mutations and transgenic lines used for the study are described in the Bloomington database [24,27]. Fly stocks were obtained from the Bloomington Stock Center, and the stock numbers are provided in Appendix A.

### 2.2. UAS/Gal4-Expression Studies and Mating Experiments

Males expressing UAS-controlled target genes were crossed to X; UAS-*dicer*; *NG4* and *OR* virgin females. Their progenies were raised at 18 °C to adulthood and shifted to 29 °C for seven days. Males and females (*X*⌃*X*, *shi ^ts^*) were then fed on apple juice agar plates at 29 °C one day prior to the mating experiment. Mating experiments were performed and mating success evaluated as previously described [24].

### 2.3. Immunofluorescence and Microscopy

Gonads were collected and stained as previously described [24]. The mouse anti-FasciclinIII (FasIII) antibody (1:10) developed by C. Goodman was obtained from the Developmental Studies Hybridoma Bank, created by the NICHD of the NIH and maintained at The University of Iowa, Department of Biology, Iowa City, IA 52242, USA. We obtained the goat anti-Vasa antibody (1:50 to 1:500) from Santa Cruz Biotechnology Inc., Dallas, TX 75220, USA (sc26877) and ordered the anti-phosphorylated Histone H3 (pHH3) antibodies from Fisher Waltman, MA 02451, USA (PA5-17869, 1:500), Milllipore, Burlington MA 01803, USA(06-570, 1:1000), and Santa Cruz Biotechnology Inc. Dallas, TX 75220, (sc8656-R, 1:100). Slow Fade Gold embedding medium with DAPI and secondary antibodies that were coupled with Alexa 488 and 568 (1:1000) were from Life Technologies, Carlsbad, CA 92008, USA. For imaging, a Zeiss Axiophot (Zeiss Microscopy, White Plains, NY 10601, USA), equipped with a digital camera, with an apotome, and Imaging software Axiovision Rel. 4.8, was used.

### 2.4. Statistical Analysis of GSc Divisions

All bar graphs and FDGs were generated using GraphPad prism version 7 and incorporated into composite images using Adobe Photoshop. Statistical relevance was analyzed using GraphPad prism default two-tailed Student’s *t*-test. GO term analysis was performed using the Gorilla (https://cbl-gorilla.cs.technion.ac.il/) and the Panther (pantherdb.org) gene ontology tools (accessed on 1 April 2025).

### 2.5. Generation of RNA-Profiles

*Oregon R* (*OR*) and *Canton S* (*CS*) were either kept in isolation (non-mated) or repeatedly mated with virgin females (mated), and male reproductive tracts were subsequently isolated. Dissected tissues were washed several times with Tissue Isolation buffer (TIB) to remove any potential contaminations from the remainder of the reproductive tract within the solution. Testes tips were cut in TIB using surgical needles and immediately transferred into Trizol reagent (#15596026, obtained from Fisher Waltman, MA 02451, USA). Total RNA was extracted using the Direct-zol RNA MiniPrep kit from Zymo Research, Irvine, CA 92614, USA, following the manufacturer’s instructions. The quality of total RNA was assayed by microcapillary electrophoresis with a Biorad BioAnalyzer (Biorad, Hercules, CA 94547, USA). Three replicates were then used for linear amplification, library construction, and sequencing (paired-end 100 bp (PE100) run on the Illumina HiSeq2000 platform (Illumina, San Diego, CA 92122, USA) by the Georgia Genomics Facility, Athens, GA 30602, USA. Resulting sequences were analyzed using TopHat2.012 and Cuffdiff2.2.1. The latter provides the statistical analysis [28,29].

## 3. Results

### 3.1. Mating Caused Differential Gene Expression

With the goal of identifying DEGs, we compared the transcriptomes of *Drosophila* testes tips from non-mated and mated males (Figure 1B), using males from two isogenized *wt* strains, *Oregon R* (*OR*), and *Canton S* (*CS*). The testis tip contains several types of cells: germline cells, somatic hub cells, somatic cyst cells, muscle cells, pigment cells, and basal membrane cells. The number of each of these cells in a *wt* testis tip is not known. However, we estimate that at least 50% of the cells are germline cells and that each *OR* testis tip used in this study contained on average 50–200 early-stage germline cells (GSCs, GBs, and SGs). The transcriptome analysis revealed 16,601 hits in the *Drosophila* database [27]. Among these were 1056 DEGs in non-mated compared to mated *OR* males and 335 DEGs in non-mated compared to mated *CS* males with *p*-values below 0.1.

Adjusting the *p*-value significantly changed the number of DEGs. The adjusted *p*-value, also called the q-value, provides the likelihood that a value is a false positive. After adjustment, we obtained a list of only 116 DEGs in *OR* and 42 DEGs in *CS*. For our analysis, this means that the genes were differentially expressed in all three non-mated to mated comparisons. Volcano blots demonstrate this difference in DEGs between the two *wt* strains (Figure 1C,D). Among the DEGs, we detected 18 transcripts (red in Appendix A) and one pseudogene (blue in Appendix A) that were differentially expressed in both *wt* strains (Figure 1E). This shows how using two different strains for this type of analysis helps to narrow down the number of DEGs. However, we noticed that some of the DEGs with a significant q-value in one *wt* strain had significantly small *p*-values in the other strain, making them a second group of candidate genes for future studies.

### 3.2. Mating Increased Expression of Secreted Proteins

Against our expectations, none of the DEGs encode transcription factors or known cell cycle regulator, based on sequence analysis and functional prediction in the *Drosophila* database. GO term analysis of the DEGs revealed that both strains had a 10-fold enrichment for secreted proteins (q-value 3 × 10^−14^). Among the 18 DEGs overlapping between the two strains, 17 are predicted to be secreted from the cells (Table 1) [27]. If the proteins encoded by the DEGs are secreted from the cells within the male gonad, they must traffic through the secretory system. Consistent with this, sequence analysis revealed potential secretion signal sequences at their N-terminal end (Table 1). The secretion signal sequence consists of a basic amino acid (red in Table 1), followed by a stretch of at least five hydrophobic amino acids (bold in Table 1) [30]. A polypeptide with a secretion signal sequence is normally recognized by a complex of proteins, the SRP, that binds to it as it exits the ribosome. The SRP is recognized and bound by the SRP-Receptor (SRPR) on the endoplasmic reticulum (ER) membrane, and this facilitates the import of the associated polypeptides into the ER [31,32,33]. As mutations in *srp* genes are lethal, we employed RNAi constructs against them from the Harvard DRSC/TRiP Functional Genomics Resources (TRiP). The RNAi hairpins are under control of the Yeast Upstream Activating Sequences (UAS), which allows for their tissue-specific expression using the UAS-GAL4 system [34]. We investigated their potential role in increasing GSC division frequency when expressed within the germline cells, using a well-established germline Gal4-transactivator, *nanos-Gal4* (*NG4*) [35]. To investigate how many of the GSCs are within mitotic divisions, we followed our standard mating and staining procedure: after mating and dissection, tissues are imaged and analyzed. GSCs are identified because they sit next to the FasIII-positive hub cells and express the germline marker, anti-Vasa (Figure 2A–C and Appendix A). Adding an antibody against pHH3 to the procedure reveals how many of the GSCs are in mitotic division.

The percentage of GSCs in mitosis is calculated by dividing the number of all pHH3-positive GSCs within a male population by the total number of GSCs within the population. This number is called the GSC M-phase index (MI^GSC^) and can be nicely demonstrated in bar graphs [22]. Normally, only a small percentage of GSCs of all GSCs within one population of males is in division. Most testes within one population have zero GSCs in division. A representative example for such testes is shown in Figure 2A and Appendix A, where nine pHH3-negative GSCs can be seen next to the hub. Significantly fewer testes from a population of mated males have zero GSCs in division. The remaining, but smaller portion of testes have mostly one (Figure 2B and Appendix A) or two (Figure 2C and Appendix A) GSCs in division. To plot the MI^GSC^ for each single testis, we use Frequency Distribution Graphs (FDGs), as shown in the Appendix A.

We discovered that expressing RNAi against *srp54* or *srp9* within the germline cells (RNAi*/NG4*) eliminated the ability of the mated males to increase their MI^GSC^ in response to mating, while control animals (RNAi*/wt*) responded normally (Figure 2D and Appendix A). Likewise, the control progeny from the *NG4* line outcrossed to *wt* increased the MI^GSC^ in response to mating (Figure 2D and Appendix A). The *NG4* line also carries a UAS-*dicer* construct (see Section 2). Loss of *dicer* affects the cell cycle in the GSCs [52]. However, neither the presence of the UAS-*dicer* construct nor its expression impaired the ability of the GSCs to increase their MI in response to mating in the present or past experiments [24]. We conclude that the secretion of proteins from the germline cells is essential for the increase in MI^GSC^ upon mating. RNAi against s*rp14* and against the *srpr* resulted in small, germline-depleted testes (Appendix A), suggesting that polypeptide import into the ER is also essential for germ cell migration or survival.

### 3.3. Accessory Glands Were Dispensable for the Increase in MI^GSC^

Many of the DEGs were originally identified from the seminal fluid which is released with the sperm during mating [42,45]. Though our transcriptome analysis was performed with only testis tips that were cut after the accessory glands were removed and the testes were thoroughly washed, we wanted to confirm that the DEGs are not required within the accessory gland for regulating MI^GSC^. Hence, we created animals without accessory glands using two different approaches. First, we destroyed secretory cells with Diphteria toxin (UAS-*dti*/*Apc26-Gal4*) and still found an increase in MI^GSC^ in mated compared to non-mated males (Appendix A). Then, we used mutant males that carry a specific allele of the segmentation gene paired (*prd^Ketel^*) which prevents the development of accessory glands [53]. As expected, males without accessory glands were sterile but still increased their MI^GSC^ upon mating (Figure 2E and Appendix A).

### 3.4. Small Proteins Were Required for the Increase in MI^GSC^ upon Mating

Among the 18 DEGs, 12 encode small proteins. To investigate if small proteins are required for the increase in MI^GSC^ in response to mating, we expressed RNAi hairpin directed against them within the germline cells using *NG4*. RNAi directed against seven of the small proteins eliminated the increase in MI^GSC^ in response to mating in experimental (RNAi*/NG4*) males but not in control (RNA*i/wt*) males (Figure 3 and Appendix A). These were *andropin* (*anp*), *activity-regulated cytoskeleton-associated protein* (*arc1/3*), *CG17242*, *bombardier* (*bbd*), *ductus ejaculatoris protein 99B* (*dup99B*), and two seminal fluid proteins (sfp), *sfp79B* and *sfp93F* (Figure 3 and Appendix A). We conclude that the expression of these small proteins is required within the germline cells for the increase in MI^GSC^ in mated males. We are yet to investigate if any of the small proteins play a role for increasing MI^GSC^ within somatic cells of the testes.

## 4. Discussion

Here, we show that small proteins regulate the frequency at which GSCs divide. The genes encoding small proteins were identified through a transcriptome analysis of testis tips from non-mated and mated *OR* and *CS* males. Using testis tips is not ideal for this analysis, because we were not using a pure population of cells. However, it appears to be the best strategy, because a GSC division is a rare process. We previously showed that we find a medium of 7% of the GSCs in division within a population of non-mated OR males [24]. To obtain statistically relevant data, we have to investigate a large number of testes (ranging from 50 to several hundred). Though laser ablation or collecting GSCs by hand would provide us with a very pure population of cells, it is not feasible to handle the large volume of required tissue for such an approach. Likewise, using single cell sequencing seems appealing, but the costs, the high volume of data, and the analysis of data associated with it would be overwhelming. To limit false positives produced by the mixed population, we used two different *wt* strains, each in triplicates for each of the conditions. While each *wt* strain produced a large set of DEGs in response to mating, adjusting the *p*-value narrowed the list of most significant changes down to only 18 genes that are likely to play a significant role in the response to mating.

We were surprised to see such a large difference (116 to 42) in DEGs between the two strains. While dissecting the males, we noted morphological differences. Testes from *OR* males were bright yellow and testes from *CS* males were orange-brown, suggesting difference in pigment production. In addition, the testes from *CS* appeared thinner than testes from *OR* males. It is possible that *CS* males had fewer early-stage germline cells at the testis tips than *OR* males, or that the germline cells were less tightly packed. It is also possible that one or the other sample contained later stage germline cells as testis tips were cut by hand, on different days, and by multiple people. A lack of consistency in the cell types could explain the differences. Besides having cellular differences in our sample, the two *wt* strains could have differences in metabolism. Environmental factors, such as nutrient availability and population density affect GSC division frequency and may also affect the response to mating [22,54]. Moreover, differences in mating behavior or success could have impacted the number of DEGs. Finally, epigenetic differences between the two strains could have impacted the level of gene transcription.

Most of the DEGs are likely secreted and, accordingly, contain a potential secretion signal sequence (Table 1) [27,51]. Consistent with a role for protein secretion in regulating MI^GSC^, reducing the expression of *srp54* and *srp9* from the germline cells led to a failure of males to increase MI^GSC^ in response to mating. Notably, a significant portion of the DEGs have previously been identified from the seminal fluid. The seminal fluid is a cocktail of bioactive molecules, including amino acids, peptides, glycoproteins, and modifying proteins that are produced in and secreted from the accessory glands [46,55]. The components of the seminal fluid are essential for sperm transfer and storage within the female body, have anti-bacterial properties, and modify female post-mating behavior [56,57,58,59,60]. We demonstrate that the accessory glands are dispensable for the increase MI^GSC^ in response to mating, suggesting roles for the DEGs within the testis.

Consistent with this, RNAi against seven of the 18 DEGs resulted in males that failed to increase MI^GSC^ in response to mating when expressed within the germline. All of these DEGs encode small proteins. Though small proteins are easy to identify, their molecular functions are harder to predict due to the lack of functional tertiary structures. The smallest protein that has been identified so far is *Drosophila* Tarsal-less (Tal). *tal* is a polycistronic gene that produces several small peptides. The smallest ones are only 11 amino acids in size and are required for development. For example, *tal* patterns the tarsal part of the *Drosophila* leg by refining the signaling border of Delta-positive and Delta-negative cells [61,62]. In tracheal, cuticular, and leg development, Tal post-transcriptionally regulates the transcription factor, Shavenbaby, by turning it from a repressor into an activator of transcription [62,63]. Another well-studied small protein is Clavata3 (CLV3). *Arabidopsis thaliana* CLV3 is 96 amino acids long and processed into an active 13-amino acid peptide. This 13-amino acid fragment of CLV3 binds to the receptor, Clavata1, and this signaling event regulates the size of the stem cell niche in the shoot meristem [64]. Small proteins have been hypothesized to act as ligands for orphan receptors. In support of this, another small protein, Apela, originally identified in zebrafish, binds to a GPCR and activates G alpha-i signaling for kidney fluid homeostasis [65]. In mice, Apela also plays a role in renal cardiac function [66].

The *Drosophila* database provides some information on the DEGs. However, the molecular and biochemical functions of most of them are not known. *anp* encodes a male-specific anti-bacterial peptide [67]. *bbd* encodes a small anti-microbial protein that acts in the Toll-mediated immune response to fight off fungi and gram-positive bacteria by regulating the levels of Bomanin peptides in the hemolymph [43]. The genome contains a cluster of three *arc* genes that are predicted to have arisen by gene duplication and modification. Among these, *arc1* plays a role in *Drosophila’s* behavioral response to metabolic stress and synaptic plasticity [38,68]. *arc3* is predicted to be a pseudogene and is almost identical to *arc1*, except that it lacks the N-terminal region. *arc2* is less conserved and has no assigned role. All three *arc* genes are similar to retroposon gag genes. Consistent with that, Arc1 and Arc2 have been shown to form capsids for the transfer of molecules at the synapse [68,69]. The RNAi hairpin for *arc* matches the conserved nucleotides of *arc1* and *arc3* and should knock down both transcripts. *CG17242* encodes a predicted protease [27]. No biochemical information is available for *sfp79B*, *sfp93F*, or *dup99B*. However, *dup99b* is homologous to the sex peptide and plays a role in oviposition. In addition to its expression in the reproductive tract, it was also detected in the hearts of both genders [47,70].

The small proteins are predicted to be secreted and their reduction from the germline cells prevented the increase in MI^GSC^ in response to mating. They could be ligands or co-factors for binding to a receptor, or part of the extra-cellular matrix that promotes cellular functions. It is possible that these small proteins bind to molecules on the germline cells, the cyst cells, or both to regulate a response to mating. All or some of them could activate some of the GPCRs required for the increase in MI^GSC^ in response to mating. Further studies should shed light on these possibilities. In addition to our finding that these small proteins are expressed at higher levels in testis tips of mated males, they were reported to be differentially expressed either in whole flies or in accessory glands after exposing flies to different stressors [45,71,72,73,74]. The response to several stressors suggests that they play multiple roles in fly development and adult tissues and that studying their roles furthers our understanding of their molecular function.

## Figures and Tables

**Figure 1 jdb-13-00021-f001:**
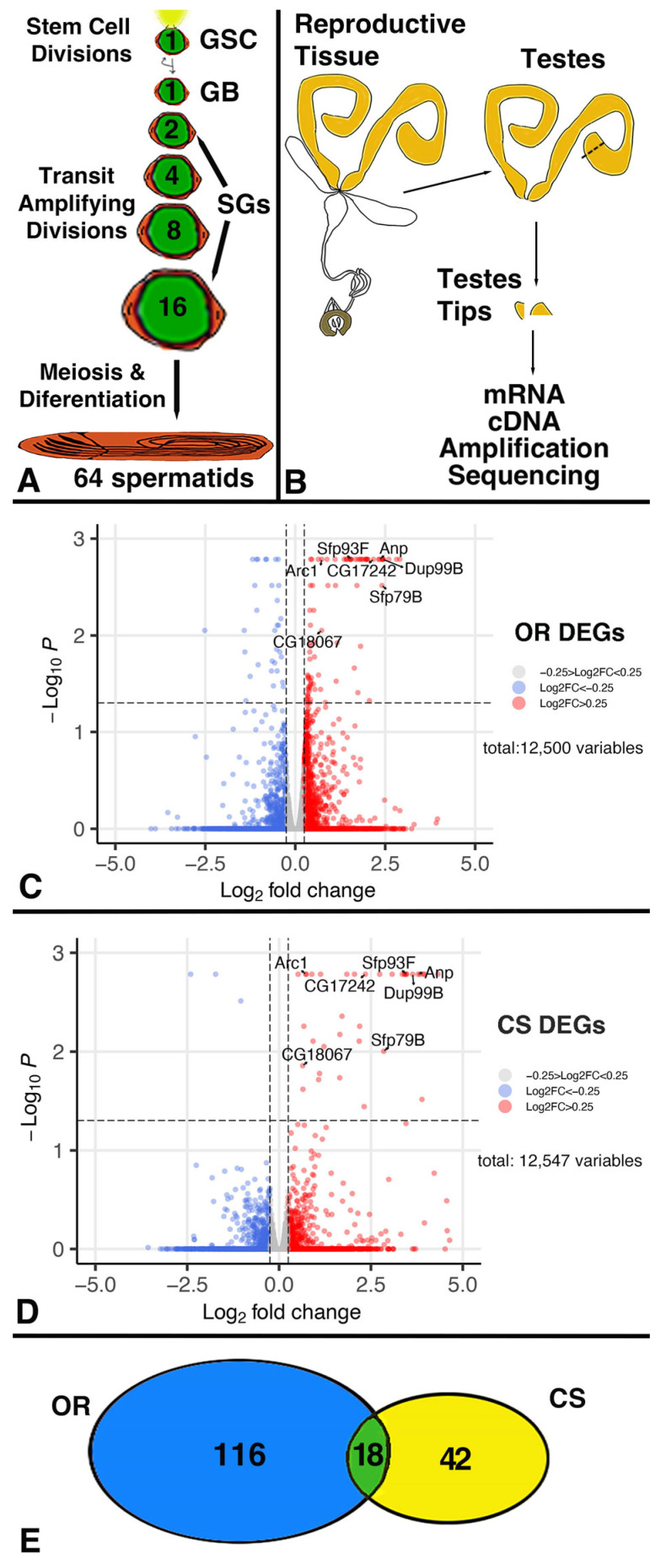
Illustrations of *Drosophila* spermatogenesis and our transcriptome analysis. (**A**) Cartoon showing how one GSC division produces 64 spermatids. GSC: Germline Stem Cell, GB: gonialblast, SG: spermatogonia. (**B**) Workflow of how DEGs were identified. (**C**,**D**) Volcano plots from *OR* and *CS*, genes as indicated. (**E**) A Venn diagram showing that 18 DEGs overlapped among the two *wt* strains.

**Figure 2 jdb-13-00021-f002:**
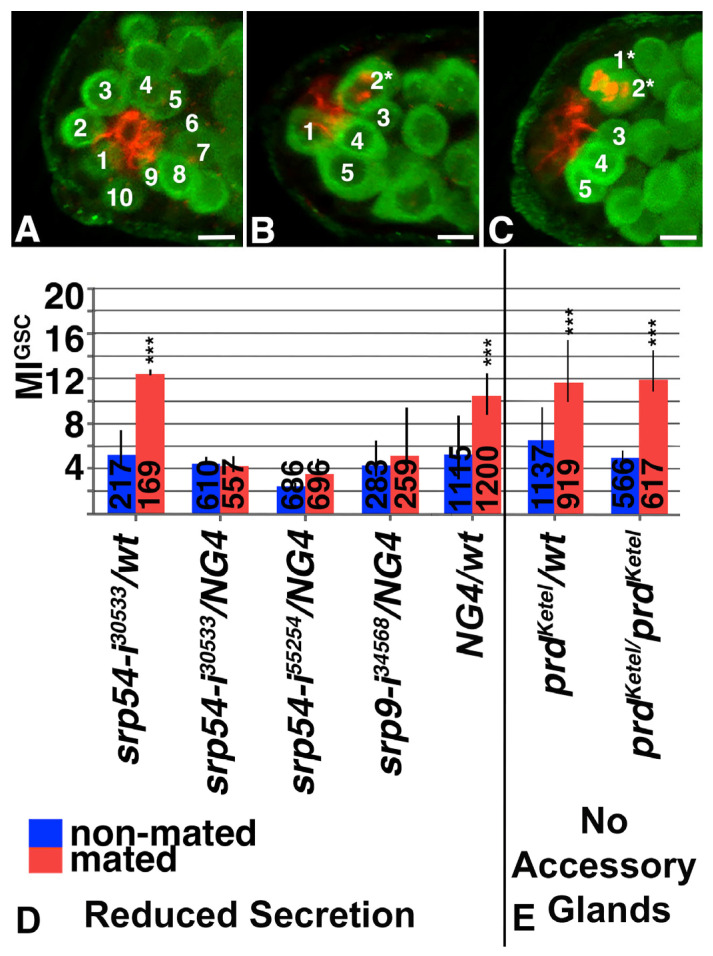
MI^GSC^ of non-mated and mated males from different genetic backgrounds. (**A**–**C**) Immuno-fluorescence images of testes tips showing Vasa-positive germline cells. GSCs around and touching the hub cells (red) are numbered. GSCs that are in mitosis (also in red) are marked by an asterisk. Scale bars represent 10 μm. Note that the testis in (**A**) shows zero out of nine GSCs in division, the testis in (**B**) shows five GSCs with one in division, and the testis in (**C**) shows five GSCs with two in division. (**D**) Males expressing RNAi against the *srp* genes failed to increase MI^GSC^ in response to mating, while control animals did. (**E**) Males without accessory glands still increased MI^GSC^ in response to mating. Genotypes: (gene abbreviation-i(for RNAi)^Bloomington stock number^) as indicated, color-coding as indicated, numbers of GSCs as indicated in each column, *** *p* < 0.001.

**Figure 3 jdb-13-00021-f003:**
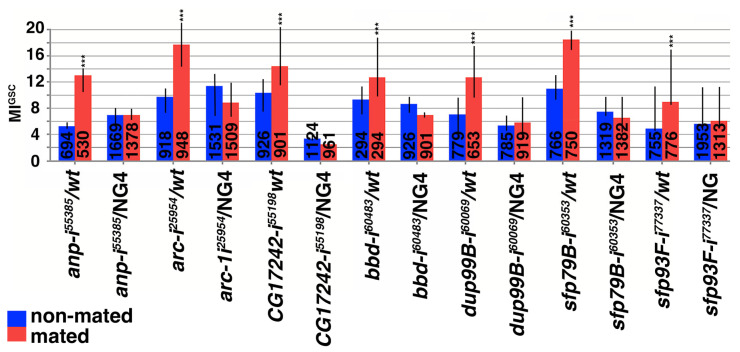
Small proteins are required for the increase in MI^GSC^. Control males and males with reduced expression of small proteins via RNAi within the germline. Genotypes: (gene abbreviation-i(for RNAi)^Bloomington stock number^) as indicated, color-coding as indicated, numbers of GSCs as indicated in each column, *** *p* < 0.001.

**Table 1 jdb-13-00021-t001:** Most DEGs carry a potential secretion signal sequence.

Gene ID	Length	Biological/Molecular Process	FlyBase Predicted Location	Potential Secretion Signal Sequence
*activity-regulated cytoskeleton associated proteins arc1/3*	254 aa 124 aa	Metabolic stress [36,37,38]	Cytoplasm	N/A
*acessory gland protein (acp) 54A1*	45 aa	Sexual reproduction [39]		MLINRHSCSK**LL**S**LMVLL**CLAFDLKPVSAM
*acessory gland protein (acp) 76A*	388 aa	Non-inhibitory Serpin [40]	extra-cellular	MGNHQV**IFLVL**CTS**LLF**QNTIQQNVSFQLI
*andropin (anp*)	57 aa	Immune response [41,42]	extra-cellular	MKY**FVVLVVL**ALILAISVGPSDAVFIDILD
*bombardier (bbd*)	250 aa	Sexual reproduction [42] Anti-microbial response, Toll signaling [43]	extra-cellular	MGSNTGAW**ILLGLLAGI**ASLSSAANIQRNE
*CG17242*	230 aa	Protease [44] Sexual Reproduction [42]	extra-cellular	MLLK**GILLLV**SIAQIAADFKSIGIEQAPWQ
*CG18258*	468 aa	Lipase [27,39]		MSSS**IAVVLVVVLIGI**SESIKTDSLMMTSS
*CG34034*	135 aa	Sexual reproduction [42,45]	extra-cellular	MSSIST**IIGL**C**LLFFML**SNVDAYGQKCSPV
*CG42521*	123 aa	Sexual reproduction [39]		MRT**VPILLLI**CCLGWLHKGQADERKIGAVG
*CG42782*	46 aa	Sexual reproduction [46]		MLISQYS**GL**K**LMLLMVGLG**MASSYEIIRQC
*CG5402*	154 aa	Sexual reproduction [42,45]	extra-cellular	MK**LLI**W**L**C**LLGFLA**SAYGIFLDKITGRGDS
*CG5162*	411 aa	Sexual reproduction [42,45] Lipase [27]	extra-cellular	MGRVPAKMHT**LLALLL**Q**LLV**ASIHAIEWSL
*ductus ejaculatorius peptide (dup) 99B*	54 aa	Sexual reproduction [42,45,47]	extra-cellular	MKTP**LFLLLVVL**ASLLGLALSQDRNDTEWI
*esterase-6 (est-6*)	500 aa	Sexual reproduction [45,48]	extra-cellular	MNY**VGLGLIIVL**SC**LWL**GSNEADPLIVEIT
*odorant binding protein (Obp) 51a*	117 aa	Sexual reproduction [42,45]	extra-cellular	MK**VFIGLVLLL**AVTTLSSALFESEANECAK
*seminal fluid protein (sfp) 79B*	35 aa	Sexual reproduction [42,45]	extra-cellular	MK**LL**S**AALVLLM**SS**ALA**M**A**QKNTNTNENNIVIGKV
*seminal fluid protein (sfp) 93F*	53 aa	Sexual reproduction [42]	extra-cellular	MLIAR**LGFLL**CS**LGLA**TAICQPNGQSCKSH
*serpin77Bb (spn77Bb*)	362 aa	Sexual reproduction [45] Hormone transport and protein folding [49,50]	extra-cellular	MK**LGFLGLFG**MVLMIMFYEGAEGYTVNELR

Gene names, protein lengths, functions, citations, and cellular locations as predicted and/or described in the database are shown in columns 1–4 [27,51]. The potential secretion signal sequences are shown in the far-right column, with hydrophobic amino acids in bold, and the charged amino acids marked in red. Note that *arc1/3* does not have a signal sequence. The sizes of the small proteins are labeled in yellow.

## Data Availability

The data are available from the authors and have been publicly deposited at GEO, accession #: GSE296024.

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
