# Peer review of "Drosophila Males Differentially Express Small Proteins Regulating Stem Cell Division Frequency in Response to Mating"

_jdb, 2025, doi:10.3390/jdb13030021_

Round 1
Reviewer 1 Report
Comments and Suggestions for Authors
The manuscript by Malpe et al. identifies new regulators involved in controlling the division rate of germline stem cells (GSCs) in response to repetitive mating. The authors present a comprehensive transcriptomic analysis of two established wild-type strains, finding that the overlapping differentially expressed genes (DEGs) largely encode secreted polypeptides.
While the current version of the manuscript appears to be experimentally well performed, the following suggestions could enhance the clarity, impact, and overall quality of the work:
Major comments:
1 - This study builds on previous work that demonstrated differences between wild-type strains in their response to repetitive mating. However, the current manuscript does not address the minimal overlap in DEGs between OR and Canton-S. The authors should provide a speculative discussion on potential genetic or epigenetic factors that could explain this discrepancy.
2 - To improve data transparency and visual clarity, the volcano plots should be enlarged. Additionally, the authors should include a comprehensive list of DEGs in a supplementary table, and highlight any genes that are downregulated in both strains.
3 - Although the evidence supporting the role of secreted peptides in modulating GSC division is compelling, the mechanism remains speculative. To strengthen the findings, the authors could perform gain-of-function experiments. For example, ectopically expressing candidate peptides in either the germline or somatic tissues could help assess their mitogenic potential.
4 - The methods section currently lacks details on the statistical tests used. A description of the statistical approach and the number of adult testes analysed per experiment is essential. While Figures 2 and 3 show GSC counts, the total number of testes analysed is unclear.
5 - Including representative images showing the differential GSC response between non-mated and mated controls and RNAi conditions would greatly help the impact of Figures 2 and 3.
6 - The current controls using OR are not ideal. The appropriate control would include UAS-Dicer flies lacking a GAL4 driver. The authors should test the effect of Dicer alone to ensure that its expression does not affect the mating-induced GSC division response.
7 - Graphs E and F in Supplementary Figure S2 appear identical. The authors should confirm whether this is an error or a coincidence, and correct if necessary.
Minor comments:
1 - The antibody list in the methods section should include specific dilutions rather than ranges to ensure reproducibility.
2 - Figures 1A and 1B are not referenced in the main text. This should be corrected to guide readers appropriately.
3 – Typographical errors:
– Page 4: "hanges" should be "changes"; "smalles" should be "smallest".
– Page 5: The first entry in the table is missing the amino acid (a.a.) information.
4 - Table S1 is labelled as showing genes upregulated in mated males, but CG42521, CG10175, CG42617, and CG43740 are downregulated. The authors should either revise the title or remove these entries.
5 - The section on accessory gland manipulations in Table S2 is not formatted in the same logical structure as the preceding sections. Harmonizing the organization would improve readability.
6 - There appears to be extraneous text on the right-hand side of Graph L in Figure S3. The authors should check and revise this figure.
Author Response
Dear Editors and Reviewer,
We are grateful for the reviewer’s comments and suggestions and are happy that the changes made in response greatly improves the quality of the manuscript. Please see our point-by-point responses below.
Major comments:
1 - This study builds on previous work that demonstrated differences between wild-type strains in their response to repetitive mating. However, the current manuscript does not address the minimal overlap in DEGs between OR and Canton-S. The authors should provide a speculative discussion on potential genetic or epigenetic factors that could explain this discrepancy.
We added a discussion why we may see these differences.
2 - To improve data transparency and visual clarity, the volcano plots should be enlarged.
We enlarged the volcano plots
Additionally, the authors should include a comprehensive list of DEGs in a supplementary table, and highlight any genes that are downregulated in both strains.
We replaced Table S1 by Tables S1 and S2 which list all DEGs with a P-value below 0.1. We like this change a lot, as it allows us to focus on the 18 DEGs with significant q-values within the main manuscript. Accordingly, we modified the venn diagram to reflect this.
3 - Although the evidence supporting the role of secreted peptides in modulating GSC division is compelling, the mechanism remains speculative. To strengthen the findings, the authors could perform gain-of-function experiments. For example, ectopically expressing candidate peptides in either the germline or somatic tissues could help assess their mitogenic potential.
We absolutely agree with the reviewer. At this point, we can only conclude that the RNAi experiment indicate a requirement for the small proteins in regulating the increase in MIGSC in response to mating. To show that they are also sufficient, we have to overexpress them, potentially even co-overexpress them. Unfortunately, the correct tools to perform these experiments in a controlled manner are currently not available. Therefore, we started generating them. Cloning, transformation into flies, establishment of fly stocks, performing crosses, and the final GSC count experiments will take us at least two years. Hence, unfortunately, those experiments are out of the scope of this manuscript.
4 - The methods section currently lacks details on the statistical tests used. A description of the statistical approach and the number of adult testes analysed per experiment is essential. While Figures 2 and 3 show GSC counts, the total number of testes analysed is unclear.
We added a statistical analysis section to the methods. The numbers of testes examined are shown in the FDGs (Figures S2 and S3). To make this clear to the reader, we added a better explanation of the numbers, and the difference between bar graphs and FDGs to the main text.
5 - Including representative images showing the differential GSC response between non-mated and mated controls and RNAi conditions would greatly help the impact of Figures 2 and 3.
We really like the idea of adding some images of tissue. It is a very difficult task to accomplish, though. Most testes do not have any GSCs in division. Upon mating, we normally find more testes that have 1, 2, or 3 cells in division. In the testes with RNAi expression, in contrast, the GSCs behave as they do in non-mated males. To improve the manuscript with images, we added images of testes tips with 0, 1, or 2 GSCs in division on top of Figure 2 and added an explanation to the text. This goes along well with our FDGs that we now explain better in the main manuscript (as in response to suggestion #4).
6 - The current controls using OR are not ideal. The appropriate control would include UAS-Dicer flies lacking a GAL4 driver. The authors should test the effect of Dicer alone to ensure that its expression does not affect the mating-induced GSC division response.
We added the second control (NG4/wt) and explain that neither the presence of the UAS-dicer insertion, nor expression of dicer within the germline cells has a noticeable effect on the increase in GSC division in response to mating.
7 - Graphs E and F in Supplementary Figure S2 appear identical. The authors should confirm whether this is an error or a coincidence, and correct if necessary.
Thank you. That was anoversight. We added the correct Graph.
Minor comments:
1 - The antibody list in the methods section should include specific dilutions rather than ranges to ensure reproducibility.
We rephrased to: Fisher (PA5-17869, 1:500), Milllipore (06-570, 1:1000), and Santa Cruz Biotechnology Inc. (sc8656-R, 1:100).
2 - Figures 1A and 1B are not referenced in the main text. This should be corrected to guide readers appropriately.
Figures 1A is referenced in the Introduction and Figure 1B is referenced at the beginning of the Results part (we highlighted)
3 – Typographical errors:
– Page 4: "hanges" should be "changes"; "smalles" should be "smallest".
– Page 5: The first entry in the table is missing the amino acid (a.a.) information.
We fixed and highlighted the typos. (We did not highlight the aa to avoid confusion with the color coding in the final version)
4 - Table S1 is labelled as showing genes upregulated in mated males, but CG42521, CG10175, CG42617, and CG43740 are downregulated. The authors should either revise the title or remove these entries.
Table S1 is now replaced by Tables S1 and S2 that show a longer list of differentially expressed genes, as requested above. However, since CG42521 is one of the DEGs with a low q-value, we changed the tile of the manuscript.
5 - The section on accessory gland manipulations in Table S2 is not formatted in the same logical structure as the preceding sections. Harmonizing the organization would improve readability.
We harmonized the organization of Table S2, which is now Table S3
6 - There appears to be extraneous text on the right-hand side of Graph L in Figure S3. The authors should check and revise this figure.
We removed the text.
Sincerely
Cordula Schulz,
Reviewer 2 Report
Comments and Suggestions for Authors
Summary:
The manuscript by Maple et al. presents findings of DEGs from teste tip tissue that are suspected to control GSC function following mating experience. The authors find several dozen genes that are differentially expressed across two independent fly strains, several of which were shown in RNAi-knockdown follow-up studies to be necessary for the GSC response.
The manuscript presents interesting findings and is generally well presented. I offer several major and minor critiques that should improve the manuscript quality. My recommendation would be for consideration of acceptance following the resubmission of a revised manuscript.
Major comments:
- RNA was collected from teste tips, as described in section 2.4. The authors should comment on the total number of cells estimated to have been used and the relative purity of cell types collected. Also, a discussion of how this preparation may influence the DEGs identified, as compared to a purer preparation or even single-cell transcriptomic approach is also encouraged.
- Do the authors know the total number of genes expressed in the GSCs that are predicted to code for secreted proteins? Data mining of any available transcriptomes of GSCs could reveal this, and a comparison to those DEGs identified herein (both regarding total number of genes, as well as the identity of these genes) could be performed and discussed. This would help put the results in a broader context for the reader.
- Scale bars should be added to Figure S1. Were images taken with the same microscope objective and at identical zoom? How were Vasa+ cell numbers counted in panels A-B, based on fluorescence intensity? A description of how the authors quantified images to conclude GSGs are lost should be added to the Figure legend or Methods.
- Was a formal analysis of GO terms for the DEGs performed? How were the “Biological/Molecular Process” terms in Table 1 determined, from interpretation of the references provided? If so, it would be valuable to also provide an unbiased analysis of DEGs by software such as DAVID to characterize functional catetories.
- The authors should provide a discussion of the possible functional connections among the 7 genes shown to be required for MI-GSC in Figure 3.
Minor comments:
- Ensure all abbreviations in Methods are described. For example, TIB in section 2.4 is not detailed.
- The authors should comment on how many of the triplicate samples were required to have shown significant expression differences of a particular gene for it to have been considered a DEG. Was this required in all 3, or perhaps 2/3?
- For a fundamental process such as gametogenesis, it is surprising the large difference in transcriptome DEGs identified in the OR and CS animals. The authors should speculate on reasons for such differences, including any potential technical issues that could have arisen. Also, are the non-overlapping genes annotated as having functions critical for reproduction?
Author Response
Dear Editors and Reviewer,
We are grateful for the reviewer’s comments and suggestions and are happy that the changes made in response greatly improves the quality of the manuscript. Please see our point-by-point responses below.
Major comments:
- RNA was collected from teste tips, as described in section 2.4. The authors should comment on the total number of cells estimated to have been used and the relative purity of cell types collected. Also, a discussion of how this preparation may influence the DEGs identified, as compared to a purer preparation or even single-cell transcriptomic approach is also encouraged.
The reviewer is making a very good point. Because we had a mixed population, we used two wt strains. Unfortunately, no one knows the number of cells at the testis tip and it most likely varies widely between genotypes and flies raised/kept under different conditions. While one could estimate the number of germline cells, cyst cells, and hub cells, no one knows how many muscle cells, pigments cells, or basal membrane cells the testes have. We are very reluctant to put a number into the manuscript because we really do not know the answer and because the field would not be happy with us if we estimate a number wrong. However, we added a discussion with an approximation in the results and discussed the technology in the discussion .
- Do the authors know the total number of genes expressed in the GSCs that are predicted to code for secreted proteins? Data mining of any available transcriptomes of GSCs could reveal this, and a comparison to those DEGs identified herein (both regarding total number of genes, as well as the identity of these genes) could be performed and discussed. This would help put the results in a broader context for the reader.
That is a good question. We do not know the total number of GSC genes predicted to code for secreted proteins. To our knowledge, there is no comprehensive transcriptome analysis of only male GSCs. There is either always a mixture of cells used in RNA sequencing or incomplete data from microarray chips. We would like to mention that we got more than 16000 hits in the database – that is literally the whole fly transcriptome. It appears that everything is expressed at the tip of the testes. We ran the GO term analysis against the whole genome, and that is the best we can do at this point their daughter cells.
- Scale bars should be added to Figure S1. Were images taken with the same microscope objective and at identical zoom? How were Vasa+ cell numbers counted in panels A-B, based on fluorescence intensity? A description of how the authors quantified images to conclude GSGs are lost should be added to the Figure legend or Methods.
Due to the changes we made to the manuscript and the supplemental data, this Figure is no longer in the manuscript. Originally, we did not show a detailed list of all DEGs and simply wanted to put this data out for anyone who may be intersted in it. Now, that we have a long list of DEGs in the supplement, this is not longer necessary and would distract from the main point of the manuscript.
- Was a formal analysis of GO terms for the DEGs performed? How were the “Biological/Molecular Process” terms in Table 1 determined, from interpretation of the references provided? If so, it would be valuable to also provide an unbiased analysis of DEGs by software such as DAVID to characterize functional catetories.
Yes, we did a GO analysis and we added the information to the mansucript (Page 5).
- The authors should provide a discussion of the possible functional connections among the 7 genes shown to be required for MI-GSC in Figure 3.
We wish we had any idea. We discussed that all showed up in our RNA seq, are small proteins, are secreted, regulate MIGSC, and could be ligands. We now added the possibility that they could regulate the seven GPCRs required for the increase in MIGSC in response to mating (last paragraph of discussion).
Minor comments:
- Ensure all abbreviations in Methods are described. For example, TIB in section 2.4 is not detailed.
We did. Thanks for pointing this out
- The authors should comment on how many of the triplicate samples were required to have shown significant expression differences of a particular gene for it to have been considered a DEG. Was this required in all 3, or perhaps 2/3?
We did. To have a q-value below 0.5, the genes have to be differentially expressed in all three replicates. Page 5
- For a fundamental process such as gametogenesis, it is surprising the large difference in transcriptome DEGs identified in the OR and CS animals. The authors should speculate on reasons for such differences, including any potential technical issues that could have arisen. Also, are the non-overlapping genes annotated as having functions critical for reproduction?
We did. Yes, there are many non-overlapping genes critical for reproduction. A full list of DEGs is now provided in Tables S1 and S2.
Sincerely
Cordula Schulz
Round 2
Reviewer 1 Report
Comments and Suggestions for Authors
The authors significantly improved the manuscript in the revised version.
Graph L in Figure S3, which still has some hanging text, is the only detail that needs to be amended before publication.
I recommend its publication in JDB.
Author Response
There appears to be extraneous text on the right-hand side of Graph L in Figure S3. The authors should check and revise this figure.
We removed the text and uploaded the correct image into the supplemental file
Reviewer 2 Report
Comments and Suggestions for Authors
The authors have adequately addressed concerns in this revised manuscript.
Author Response
Thank you for the positive feedback